# The Fast Convergence of Incremental PCA

**Akshay Balsubramani**
UC San Diego
abalsubr@cs.ucsd.edu

**Sanjoy Dasgupta**
UC San Diego
dasgupta@cs.ucsd.edu

**Yoav Freund**
UC San Diego
yfreund@cs.ucsd.edu

## Abstract

We consider a situation in which we see samples $X_n \in \mathbb{R}^d$ drawn i.i.d. from some distribution with mean zero and unknown covariance $A$. We wish to compute the top eigenvector of $A$ in an incremental fashion - with an algorithm that maintains an estimate of the top eigenvector in $O(d)$ space, and incrementally adjusts the estimate with each new data point that arrives. Two classical such schemes are due to Krasulina (1969) and Oja (1983). We give finite-sample convergence rates for both.

## 1   Introduction

Principal component analysis (PCA) is a popular form of dimensionality reduction that projects a data set on the top eigenvector(s) of its covariance matrix. The default method for computing these eigenvectors uses $O(d^2)$ space for data in $\mathbb{R}^d$, which can be prohibitive in practice. It is therefore of interest to study incremental schemes that take one data point at a time, updating their estimates of the desired eigenvectors with each new point. For computing one eigenvector, such methods use $O(d)$ space.

For the case of the top eigenvector, this problem has long been studied, and two elegant solutions were obtained by Krasulina [7] and Oja [9]. Their methods are closely related. At time $n-1$, they have some estimate $V_{n-1} \in \mathbb{R}^d$ of the top eigenvector. Upon seeing the next data point, $X_n$, they update this estimate as follows:

$$V_n = V_{n-1} + \gamma_n \left( X_n X_n^T - \frac{V_{n-1}^T X_n X_n^T V_{n-1}}{\|V_{n-1}\|^2} I_d \right) V_{n-1} \qquad \text{(Krasulina)}$$

$$V_n = \frac{V_{n-1} + \gamma_n X_n X_n^T V_{n-1}}{\|V_{n-1} + \gamma_n X_n X_n^T V_{n-1}\|} \qquad \text{(Oja)}$$

Here $\gamma_n$ is a "learning rate" that is typically proportional to $1/n$.

Suppose the points $X_1, X_2, \ldots$ are drawn i.i.d. from a distribution on $\mathbb{R}^d$ with mean zero and covariance matrix $A$. The original papers proved that these estimators converge almost surely to the top eigenvector of $A$ (call it $v^*$) under mild conditions:

- $\sum_n \gamma_n = \infty$ while $\sum_n \gamma_n^2 < \infty$.
- If $\lambda_1, \lambda_2$ denote the top two eigenvalues of $A$, then $\lambda_1 > \lambda_2$.
- $\mathbb{E}\|X_n\|^k < \infty$ for some suitable $k$ (for instance, $k = 8$ works).

There are also other incremental estimators for which convergence has not been established; see, for instance, [12] and [16].

In this paper, we analyze the rate of convergence of the Krasulina and Oja estimators. They can be treated in a common framework, as stochastic approximation algorithms for maximizing the

Rayleigh quotient

$$G(v) = \frac{v^T A v}{v^T v}.$$

The maximum value of this function is $\lambda_1$, and is achieved at $v^*$ (or any nonzero multiple thereof). The gradient is

$$\nabla G(v) = \frac{2}{\|v\|^2} \left( A - \frac{v^T A v}{v^T v} I_d \right) v.$$

Since $\mathbb{E} X_n X_n^T = A$, we see that Krasulina's method is stochastic gradient descent. The Oja procedure is closely related: as pointed out in [10], the two are identical to within second-order terms.

Recently, there has been a lot of work on rates of convergence for stochastic gradient descent (for instance, [11]), but this has typically been limited to convex cost functions. These results do not apply to the non-convex Rayleigh quotient, except at the very end, when the system is near convergence. Most of our analysis focuses on the buildup to this finale.

We measure the quality of the solution $V_n$ at time $n$ using the potential function

$$\Psi_n = 1 - \frac{(V_n \cdot v^*)^2}{\|V_n\|^2},$$

where $v^*$ is taken to have unit norm. This quantity lies in the range $[0, 1]$, and we are interested in the rate at which it approaches zero. The result, in brief, is that $\mathbb{E}[\Psi_n] = O(1/n)$, under conditions that are similar to those above, but stronger. In particular, we require that $\gamma_n$ be proportional to $1/n$ and that $\|X_n\|$ be bounded.

## 1.1 The algorithm

We analyze the following procedure.

1. **Set starting time.** Set the clock to time $n_o$.
2. **Initialization.** Initialize $V_{n_o}$ uniformly at random from the unit sphere in $\mathbb{R}^d$.
3. For time $n = n_o + 1, n_o + 2, \ldots$:
   (a) Receive the next data point, $X_n$.
   (b) **Update step.** Perform either the Krasulina or Oja update, with $\gamma_n = c/n$.

The first step is similar to using a learning rate of the form $\gamma_n = c/(n + n_o)$, as is often done in stochastic gradient descent implementations [1]. We have adopted it because the initial sequence of updates is highly noisy: during this phase $V_n$ moves around wildly, and cannot be shown to make progress. It becomes better behaved when the step size $\gamma_n$ becomes smaller, that is to say when $n$ gets larger than some suitable $n_o$. By setting the start time to $n_o$, we can simply fast-forward the analysis to this moment.

## 1.2 Initialization

One possible initialization is to set $V_{n_o}$ to the first data point that arrives, or to the average of a few data points. This seems sensible enough, but can fail dramatically in some situations.

Here is an example. Suppose $X$ can take on just $2d$ possible values: $\pm e_1, \pm \sigma e_2, \ldots, \pm \sigma e_d$, where the $e_i$ are coordinate directions and $0 < \sigma < 1$ is a small constant. Suppose further that the distribution of $X$ is specified by a single positive number $p < 1$:

$$\Pr(X = e_1) = \Pr(X = -e_1) = \frac{p}{2}$$

$$\Pr(X = \sigma e_i) = \Pr(X = -\sigma e_i) = \frac{1-p}{2(d-1)} \quad \text{for } i > 1$$

Then $X$ has mean zero and covariance $\mathrm{diag}(p, \sigma^2(1-p)/(d-1), \ldots, \sigma^2(1-p)/(d-1))$. We will assume that $p$ and $\sigma$ are chosen so that $p > \sigma^2(1-p)/(d-1)$; in our notation, the top eigenvalues are then $\lambda_1 = p$ and $\lambda_2 = \sigma^2(1-p)/(d-1)$, and the target vector is $v^* = e_1$.

If $V_n$ is ever orthogonal to some $e_i$, it will remain so forever. This is because both the Krasulina and Oja updates have the following properties:

$$V_{n-1} \cdot X_n = 0 \quad \Longrightarrow \quad V_n = V_{n-1}$$
$$V_{n-1} \cdot X_n \neq 0 \quad \Longrightarrow \quad V_n \in \text{span}(V_{n-1}, X_n).$$

If $V_{n_o}$ is initialized to a random data point, then with probability $1 - p$, it will be assigned to some $e_i$ with $i > 1$, and will converge to a multiple of that same $e_i$ rather than to $e_1$. Likewise, if it is initialized to the average of $\leq 1/p$ data points, then with constant probability it will be orthogonal to $e_1$ and remain so always.

Setting $V_{n_o}$ to a random unit vector avoids this problem. However, there are doubtless cases, for instance when the data has intrinsic dimension $\ll d$, in which a better initializer is possible.

### 1.3 The setting of the learning rate

In order to get a sense of what rates of convergence we might expect, let's return to the example of a random vector $X$ with $2d$ possible values. In the Oja update $V_n = V_{n-1} + \gamma_n X_n X_n^T V_{n-1}$, we can ignore normalization if we are merely interested in the progress of the potential function $\Psi_n$. Since the $X_n$ correspond to coordinate directions, each update changes just one coordinate of $V$:

$$X_n = \pm e_1 \quad \Longrightarrow \quad V_{n,1} = V_{n-1,1}(1 + \gamma_n)$$
$$X_n = \pm \sigma e_i \quad \Longrightarrow \quad V_{n,i} = V_{n-1,i}(1 + \sigma^2 \gamma_n)$$

Recall that we initialize $V_{n_o}$ to a random vector from the unit sphere. For simplicity, let's just suppose that $n_o = 0$ and that this initial value is the all-ones vector (again, we don't have to worry about normalization). On each iteration the first coordinate is updated with probability exactly $p = \lambda_1$, and thus

$$\mathbb{E}[V_{n,1}] = (1 + \lambda_1 \gamma_1)(1 + \lambda_1 \gamma_2) \cdots (1 + \lambda_1 \gamma_n) \sim \exp(\lambda_1(\gamma_1 + \cdots + \gamma_n)) \sim n^{c\lambda_1}$$

since $\gamma_n = c/n$. Likewise, for $i > 1$,

$$\mathbb{E}[V_{n,i}] = (1 + \lambda_2 \gamma_1)(1 + \lambda_2 \gamma_2) \cdots (1 + \lambda_2 \gamma_n) \sim n^{c\lambda_2}.$$

If all goes according to expectation, then at time $n$,

$$\Psi_n = 1 - \frac{V_{n,1}^2}{\|V_n\|^2} \sim 1 - \frac{n^{2c\lambda_1}}{n^{2c\lambda_1} + (d-1)n^{2c\lambda_2}} \sim \frac{d-1}{n^{2c(\lambda_1 - \lambda_2)}}.$$

(This is all very rough, but can be made precise by obtaining concentration bounds for $\ln V_{n,i}$.) From this, we can see that it is not possible to achieve a $O(1/n)$ rate unless $c \geq 1/(2(\lambda_1 - \lambda_2))$. Therefore, we will assume this when stating our final results, although most of our analysis is in terms of general $\gamma_n$. An interesting practical question, to which we do not have an answer, is how one would empirically set $c$ without prior knowledge of the eigenvalue gap.

### 1.4 Nested sample spaces

For $n \geq n_o$, let $\mathcal{F}_n$ denote the sigma-field of all outcomes up to and including time $n$: $\mathcal{F}_n = \sigma(V_{n_o}, X_{n_o+1}, \ldots, X_n)$. We start by showing that

$$\mathbb{E}[\Psi_n | \mathcal{F}_{n-1}] \leq \Psi_{n-1}(1 - 2\gamma_n(\lambda_1 - \lambda_2)(1 - \Psi_{n-1})) + O(\gamma_n^2).$$

Initially $\Psi_n$ is likely to be close to 1. For instance, if the initial $V_{n_o}$ is picked uniformly at random from the surface of the unit sphere in $\mathbb{R}^d$, then we'd expect $\Psi_{n_o} \approx 1 - 1/d$. This means that the initial rate of decrease is very small, because of the $(1 - \Psi_{n-1})$ term.

To deal with this, we divide the analysis into epochs: the first takes $\Psi_n$ from $1 - 1/d$ to $1 - 2/d$, the second from $1 - 2/d$ to $1 - 4/d$, and so on until $\Psi_n$ finally drops below $1/2$. We use martingale large deviation bounds to bound the length of each epoch, and also to argue that $\Psi_n$ does not regress. In particular, we establish a sequence of times $n_j$ such that (with high probability)

$$\sup_{n \geq n_j} \Psi_n \leq 1 - \frac{2^j}{d}. \tag{1}$$

The analysis of each epoch uses martingale arguments, but at the same time, assumes that $\Psi_n$ remains bounded above. Combining the two requires a careful specification of the sample space at each step. Let $\Omega$ denote the sample space of all realizations $(v_{n_o}, x_{n_o+1}, x_{n_o+2}, \ldots)$, and $P$ the probability distribution on these sequences. For any $\delta > 0$, we define a nested sequence of spaces $\Omega \supset \Omega'_{n_o} \supset \Omega'_{n_o+1} \supset \cdots$ such that each $\Omega'_n$ is $\mathcal{F}_{n-1}$-measurable, has probability $P(\Omega'_n) \geq 1 - \delta$, and moreover consists exclusively of realizations $\omega \in \Omega$ that satisfy the constraints (1) up to and including time $n - 1$. We can then build martingale arguments by restricting attention to $\Omega'_n$ when computing the conditional expectations of quantities at time $n$.

## 1.5  Main result

We make the following assumptions:

(A1) The $X_n \in \mathbb{R}^d$ are i.i.d. with mean zero and covariance $A$.

(A2) There is a constant $B$ such that $\|X_n\|^2 \leq B$.

(A3) The eigenvalues $\lambda_1 \geq \lambda_2 \geq \cdots \geq \lambda_d$ of $A$ satisfy $\lambda_1 > \lambda_2$.

(A4) The step sizes are of the form $\gamma_n = c/n$.

Under these conditions, we get the following rate of convergence for the Krasulina update.

**Theorem 1.1.** *There are absolute constants $A_o, A_1 > 0$ and $1 < a < 4$ for which the following holds. Pick any $0 < \delta < 1$, and any $c_o > 2$. Set the step sizes to $\gamma_n = c/n$, where $c = c_o/(2(\lambda_1 - \lambda_2))$, and set the starting time to $n_o \geq (A_o B^2 c^2 d^2 / \delta^4) \ln(1/\delta)$. Then there is a nested sequence of subsets of the sample space $\Omega \supset \Omega'_{n_o} \supset \Omega'_{n_o+1} \supset \cdots$ such that for any $n \geq n_o$, we have:*

$$P(\Omega'_n) \geq 1 - \delta \ \ and$$

$$\mathbb{E}_n \left[ \frac{(V_n \cdot v^*)^2}{\|V_n\|^2} \right] \geq 1 - \left( \frac{c^2 B^2 e^{c_o/n_o}}{2(c_o - 2)} \right) \frac{1}{n+1} - A_1 \left( \frac{d}{\delta^2} \right)^a \left( \frac{n_o + 1}{n + 1} \right)^{c_o/2},$$

*where $\mathbb{E}_n$ denotes expectation restricted to $\Omega'_n$.*

Since $c_o > 2$, this bound is of the form $\mathbb{E}_n[\Psi_n] = O(1/n)$.

The result above also holds for the Oja update up to absolute constants.

We also remark that a small modification to the final step in the proof of the above yields a rate of $\mathbb{E}_n[\Psi_n] = O(n^{-a})$, with an identical definition of $\mathbb{E}_n[\Psi_n]$. The details are in the proof, in Appendix D.2.

## 1.6  Related work

There is an extensive line of work analyzing PCA from the statistical perspective, in which the convergence of various estimators is characterized under certain conditions, including generative models of the data [5] and various assumptions on the covariance matrix spectrum [14, 4] and eigenvalue spacing [17]. Such works do provide finite-sample guarantees, but they apply only to the batch case and/or are computationally intensive, rather than considering an efficient incremental algorithm.

Among incremental algorithms, the work of Warmuth and Kuzmin [15] describes and analyzes worst-case online PCA, using an experts-setting algorithm with a super-quadratic per-iteration cost. More efficient general-purpose incremental PCA algorithms have lacked finite-sample analyses [2]. There have been recent attempts to remedy this situation by relaxing the nonconvexity inherent in the problem [3] or making generative assumptions [8]. The present paper directly analyzes the oldest known incremental PCA algorithms under relatively mild assumptions.

## 2  Outline of proof

We now sketch the proof of Theorem 1.1; almost all the details are relegated to the appendix.

Recall that for $n \geq n_o$, we take $\mathcal{F}_n$ to be the sigma-field of all outcomes up to and including time $n$, that is, $\mathcal{F}_n = \sigma(V_{n_o}, X_{n_o+1}, \ldots, X_n)$.

An additional piece of notation: we will use $\widehat{u}$ to denote $u/\|u\|$, the unit vector in the direction of $u \in \mathbb{R}^d$. Thus, for instance, the Rayleigh quotient can be written $G(v) = \widehat{v}^T A \widehat{v}$.

## 2.1 Expected per-step change in potential

We first bound the expected improvement in $\Psi_n$ in each step of the Krasulina or Oja algorithms.

**Theorem 2.1.** *For any $n > n_o$, we can write $\Psi_n \leq \Psi_{n-1} + \beta_n - Z_n$, where*

$$\beta_n = \begin{cases} \gamma_n^2 B^2/4 & \text{(Krasulina)} \\ 5\gamma_n^2 B^2 + 2\gamma_n^3 B^3 & \text{(Oja)} \end{cases}$$

*and where $Z_n$ is a $\mathcal{F}_n$-measurable random variable with the following properties:*

- $\mathbb{E}[Z_n|\mathcal{F}_{n-1}] = 2\gamma_n(\widehat{V}_{n-1} \cdot v^*)^2(\lambda_1 - G(V_{n-1})) \geq 2\gamma_n(\lambda_1 - \lambda_2)\Psi_{n-1}(1 - \Psi_{n-1}) \geq 0$.

- $|Z_n| \leq 4\gamma_n B$.

The theorem follows from Lemmas **??** and **??** in the appendix. Its characterization of the two estimators is almost identical, and for simplicity we will henceforth deal only with Krasulina's estimator. All the subsequent results hold also for Oja's method, up to constants.

## 2.2 A large deviation bound for $\Psi_n$

We know from Theorem 2.1 that $\Psi_n \leq \Psi_{n-1} + \beta_n - Z_n$, where $\beta_n$ is non-stochastic and $Z_n$ is a quantity of positive expected value. Thus, in expectation, and modulo a small additive term, $\Psi_n$ decreases monotonically. However, the amount of decrease at the $n$th time step can be arbitrarily small when $\Psi_n$ is close to 1. Thus, we need to show that $\Psi_n$ is eventually bounded away from 1, i.e. there exists some $\epsilon_o > 0$ and some time $n_o$ such that for any $n \geq n_o$, we have $\Psi_n \leq 1 - \epsilon_o$.

Recall from the algorithm specification that we advance the clock so as to skip the pre-$n_o$ phase. Given this, what can we expect $\epsilon_o$ to be? If the initial estimate $V_{n_o}$ is a random unit vector, then $\mathbb{E}[\Psi_{n_o}] = 1 - 1/d$ and, roughly speaking, $\Pr(\Psi_{n_o} > 1 - \epsilon/d) = O(\sqrt{\epsilon})$. If $n_o$ is sufficiently large, then $\Psi_n$ may subsequently increase a little bit, but not by very much. In this section, we establish the following bound.

**Theorem 2.2.** *Suppose the initial estimate $V_{n_o}$ is chosen uniformly at random from the surface of the unit sphere in $\mathbb{R}^d$. Assume also that the step sizes are of the form $\gamma_n = c/n$, for some constant $c > 0$. Then for any $0 < \epsilon < 1$, if $n_o \geq 2B^2c^2d^2/\epsilon^2$, we have*

$$\Pr\left(\sup_{n \geq n_o} \Psi_n \geq 1 - \frac{\epsilon}{d}\right) \leq \sqrt{2e\epsilon}.$$

To prove this, we start with a simple recurrence for the moment-generating function of $\Psi_n$.

**Lemma 2.3.** *Consider a filtration $(\mathcal{F}_n)$ and random variables $Y_n, Z_n \in \mathcal{F}_n$ such that there are two sequences of nonnegative constants, $(\beta_n)$ and $(\zeta_n)$, for which:*

- $Y_n \leq Y_{n-1} + \beta_n - Z_n$.

- *Each $Z_n$ takes values in an interval of length $\zeta_n$.*

*Then for any $t > 0$, we have $\mathbb{E}[e^{tY_n}|\mathcal{F}_{n-1}] \leq \exp(t(Y_{n-1} - \mathbb{E}[Z_n|\mathcal{F}_{n-1}] + \beta_n + t\zeta_n^2/8))$.*

This relation shows how to define a supermartingale based on $e^{tY_n}$, from which we can derive a large deviation bound on $Y_n$.

**Lemma 2.4.** *Assume the conditions of Lemma 2.3, and also that $\mathbb{E}[Z_n|\mathcal{F}_{n-1}] \geq 0$. Then, for any integer $m$ and any $\Delta, t > 0$,*

$$\Pr\left(\sup_{n \geq m} Y_n \geq \Delta\right) \leq \mathbb{E}[e^{tY_m}] \exp\left(-t\left(\Delta - \sum_{\ell > m}(\beta_\ell + t\zeta_\ell^2/8)\right)\right).$$

In order to apply this to the sequence $(\Psi_n)$, we need to first calculate the moment-generating function of its starting value $\Psi_{n_o}$.

**Lemma 2.5.** *Suppose a vector $V$ is picked uniformly at random from the surface of the unit sphere in $\mathbb{R}^d$, where $d \geq 3$. Define $Y = 1 - (V_1^2)/\|V\|^2$. Then, for any $t > 0$,*

$$\mathbb{E}e^{tY} \leq e^t \sqrt{\frac{d-1}{2t}}.$$

Putting these pieces together yields Theorem 2.2.

## 2.3 Intermediate epochs of improvement

We have seen that, for suitable $\epsilon$ and $n_o$, it is likely that $\Psi_n \leq 1 - \epsilon/d$ for all $n \geq n_o$. We now define a series of epochs in which $1 - \Psi_n$ successively doubles, until $\Psi_n$ finally drops below $1/2$.

To do this, we specify intermediate goals $(n_o, \epsilon_o), (n_1, \epsilon_1), (n_2, \epsilon_2), \ldots, (n_J, \epsilon_J)$, where $n_o < n_1 < \cdots < n_J$ and $\epsilon_o < \epsilon_1 < \cdots < \epsilon_J = 1/2$, with the intention that:

$$\text{For all } 0 \leq j \leq J, \text{ we have } \sup_{n \geq n_j} \Psi_n \leq 1 - \epsilon_j. \tag{2}$$

Of course, this can only hold with a certain probability.

Let $\Omega$ denote the sample space of all realizations $(v_{n_o}, x_{n_o+1}, x_{n_o+2}, \ldots)$, and $P$ the probability distribution on these sequences. We will show that, for a certain choice of $\{(n_j, \epsilon_j)\}$, all $J+1$ constraints (2) can be met by excluding just a small portion of $\Omega$.

We consider a specific realization $\omega \in \Omega$ to be good if it satisfies (2). Call this set $\Omega'$:

$$\Omega' = \{\omega \in \Omega : \sup_{n \geq n_j} \Psi_n(\omega) \leq 1 - \epsilon_j \text{ for all } 0 \leq j \leq J\}.$$

For technical reasons, we also need to look at realizations that are good up to time $n-1$. Specifically, for each $n$, define

$$\Omega'_n = \{\omega \in \Omega : \sup_{n_j \leq \ell < n} \Psi_\ell(\omega) \leq 1 - \epsilon_j \text{ for all } 0 \leq j \leq J\}.$$

Crucially, this is $\mathcal{F}_{n-1}$-measurable. Also note that $\Omega' = \bigcap_{n > n_o} \Omega'_n$.

We can talk about expectations under the distribution $P$ restricted to subsets of $\Omega$. In particular, let $P_n$ be the restriction of $P$ to $\Omega'_n$; that is, for any $A \subset \Omega$, we have $P_n(A) = P(A \cap \Omega'_n)/P(\Omega'_n)$. As for expectations with respect to $P_n$, for any function $f : \Omega \to \mathbb{R}$, we define

$$\mathbb{E}_n f = \frac{1}{P(\Omega'_n)} \int_{\Omega'_n} f(\omega) P(d\omega).$$

Here is the main result of this section.

**Theorem 2.6.** *Assume that $\gamma_n = c/n$, where $c = c_o/(2(\lambda_1 - \lambda_2))$ and $c_o > 0$. Pick any $0 < \delta < 1$ and select a schedule $(n_o, \epsilon_o), \ldots, (n_J, \epsilon_J)$ that satisfies the conditions*

$$\epsilon_o = \frac{\delta^2}{8ed}, \text{ and } \frac{3}{2}\epsilon_j \leq \epsilon_{j+1} \leq 2\epsilon_j \text{ for } 0 \leq j < J, \text{ and } \epsilon_{J-1} \leq \frac{1}{4}$$

$$(n_{j+1} + 1) \geq e^{5/c_o}(n_j + 1) \text{ for } 0 \leq j < J \tag{3}$$

*as well as $n_o \geq (20c^2 B^2/\epsilon_o^2) \ln(4/\delta)$. Then $\Pr(\Omega') \geq 1 - \delta$.*

The first step towards proving this theorem is bounding the moment-generating function of $\Psi_n$ in terms of that of $\Psi_{n-1}$.

**Lemma 2.7.** *Suppose $n > n_j$. Suppose also that $\gamma_n = c/n$, where $c = c_o/(2(\lambda_1 - \lambda_2))$. Then for any $t > 0$,*

$$\mathbb{E}_n[e^{t\Psi_n}] \leq \mathbb{E}_n \left[\exp\left(t\Psi_{n-1}\left(1 - \frac{c_o\epsilon_j}{n}\right)\right)\right] \exp\left(\frac{c^2 B^2 t(1+32t)}{4n^2}\right).$$

We would like to use this result to bound $\mathbb{E}_n[\Psi_n]$ in terms of $\mathbb{E}_m[\Psi_m]$ for $m < n$. The shift in sample spaces is easily handled using the following observation.

**Lemma 2.8.** *If $g : \mathbb{R} \to \mathbb{R}$ is nondecreasing, then $\mathbb{E}_n[g(\Psi_{n-1})] \leq \mathbb{E}_{n-1}[g(\Psi_{n-1})]$ for any $n > n_o$.*

A repeated application of Lemmas 2.7 and 2.8 yields the following.

**Lemma 2.9.** *Suppose that conditions (3) hold. Then for $0 \leq j < J$ and any $t > 0$,*

$$\mathbb{E}_{n_{j+1}}[e^{t\Psi_{n_{j+1}}}] \leq \exp\left(t(1 - \epsilon_{j+1}) - t\epsilon_j + \frac{tc^2 B^2(1 + 32t)}{4}\left(\frac{1}{n_j} - \frac{1}{n_{j+1}}\right)\right).$$

Now that we have bounds on the moment-generating functions of intermediate $\Psi_n$, we can apply martingale deviation bounds, as in Lemma 2.4, to obtain the following, from which Theorem 2.6 ensues.

**Lemma 2.10.** *Assume conditions (3) hold. Pick any $0 < \delta < 1$, and set $n_o \geq (20c^2 B^2/\epsilon_o^2)\ln(4/\delta)$. Then*

$$\sum_{j=1}^{J} P_{n_j}\left(\sup_{n \geq n_j} \Psi_n > 1 - \epsilon_j\right) \leq \frac{\delta}{2}.$$

### 2.4 The final epoch

Recall the definition of the intermediate goals $(n_j, \epsilon_j)$ in (2), (3). The final epoch is the period $n \geq n_J$, at which point $\Psi_n \leq 1/2$. The following consequence of Lemmas **??** and 2.8 captures the rate at which $\Psi$ decreases during this phase.

**Lemma 2.11.** *For all $n > n_J$,*

$$\mathbb{E}_n[\Psi_n] \leq (1 - \alpha_n)\mathbb{E}_{n-1}[\Psi_{n-1}] + \beta_n,$$

*where $\alpha_n = (\lambda_1 - \lambda_2)\gamma_n$ and $\beta_n = (B^2/4)\gamma_n^2$.*

By solving this recurrence relation, and piecing together the various epochs, we get the overall convergence result of Theorem 1.1.

Note that Lemma 2.11 closely resembles the recurrence relation followed by the squared $L^2$ distance from the optimum of stochastic gradient descent (SGD) on a strongly convex function [11]. As $\Psi_n \to 0$, the incremental PCA algorithms we study have convergence rates of the same form as SGD in this scenario.

## 3 Experiments

When performing PCA in practice with massive $d$ and a large/growing dataset, an incremental method like that of Krasulina or Oja remains practically viable, even as quadratic-time and -memory algorithms become increasingly impractical. Arora et al. [2] have a more complete discussion of the empirical necessity of incremental PCA algorithms, including a version of Oja's method which is shown to be extremely competitive in practice.

Since the efficiency benefits of these types of algorithms are well understood, we now instead focus on the effect of the learning rate on the performance of Oja's algorithm (results for Krasulina's are extremely similar). We use the CMU PIE faces [13], consisting of 11554 images of size $32 \times 32$, as a prototypical example of a dataset with most of its variance captured by a few PCs, as shown in Fig. 1. We set $n_0 = 0$.

We expect from Theorem 1.1 and the discussion in the introduction that varying $c$ (the constant in the learning rate) will influence the overall rate of convergence. In particular, if $c$ is low, then halving it can be expected to halve the exponent of $n$, and the slope of the log-log convergence graph (ref. the remark after Thm. 1.1). This is exactly what occurs in practice, as illustrated in Fig. 2. The dotted line in that figure is a convergence rate of $1/n$, drawn as a guide.

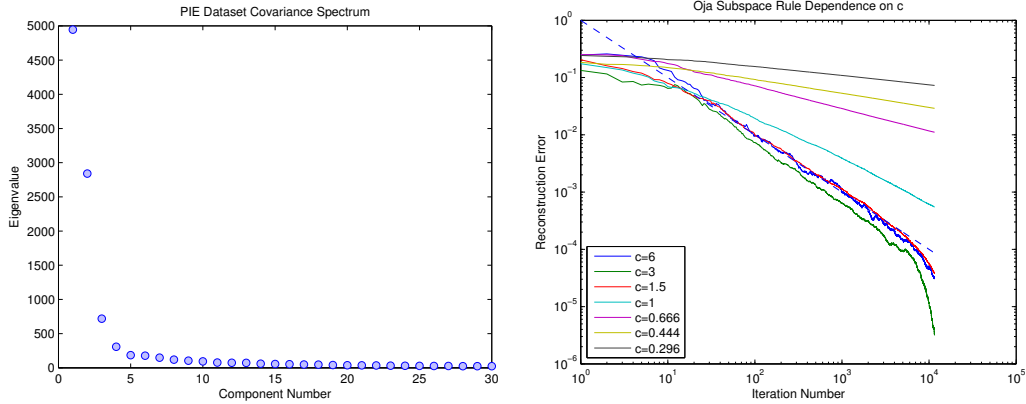

Figures 1 and 2.

## 4 Open problems

Several fundamental questions remain unanswered. First, the convergence rates of the two incremental schemes depend on the multiplier $c$ in the learning rate $\gamma_n$. If it is too low, convergence will be slower than $O(1/n)$. If it is too high, the constant in the rate of convergence will be large. Is there a simple and practical scheme for setting $c$?

Second, what can be said about incrementally estimating the top $p$ eigenvectors, for $p > 1$? Both methods we consider extend easily to this case [10]; the estimate at time $n$ is a $d \times p$ matrix $V_n$ whose columns correspond to the eigenvectors, with the invariant $V_n^T V_n = I_p$ always maintained. In Oja's algorithm, for instance, when a new data point $X_n \in \mathbb{R}^d$ arrives, the following update is performed:

$$W_n = V_{n-1} + \gamma_n X_n X_n^T V_{n-1}$$
$$V_n = \text{orth}(W_n)$$

where the second step orthonormalizes the columns, for instance by Gram-Schmidt. It would be interesting to characterize the rate of convergence of this scheme.

Finally, our analysis applies to a modified procedure in which the starting time $n_o$ is artificially set to a large constant. This seems unnecessary in practice, and it would be useful to extend the analysis to the case where $n_o = 0$.

### Acknowledgments

The authors are grateful to the National Science Foundation for support under grant IIS-1162581.

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
