[Supplementary Material]

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

# A Expected per-step change in potential

## A.1 The change in potential of Krasulina's update

Write Krasulina's update equation as

$$V_n = V_{n-1} + \gamma_n \xi_n$$

$$\xi_n = \left( X_n X_n^T - \widehat{V}_{n-1}^T X_n X_n^T \widehat{V}_{n-1} I_d \right) V_{n-1}$$

We start with some basic observations.

**Lemma A.1.** *For all $n > n_o$,*

    *(a) $\xi_n$ is orthogonal to $V_{n-1}$.*

    *(b) $\|\xi_n\|^2 \le B^2 \|V_{n-1}\|^2 / 4$.*

    *(c) $\mathbb{E}[\xi_n | \mathcal{F}_{n-1}] = A V_{n-1} - G(V_{n-1}) V_{n-1}$.*

    *(d) $\|V_n\| \ge \|V_{n-1}\|$.*

*Proof.* For (a), let $X_n^{\perp}$ denote the component of $X_n$ orthogonal to $V_{n-1}$. Then

$$\xi_n = (V_{n-1} \cdot X_n) X_n - (\widehat{V}_{n-1} \cdot X_n)^2 V_{n-1} = (V_{n-1} \cdot X_n)(X_n - (\widehat{V}_{n-1} \cdot X_n)\widehat{V}_{n-1}) = (V_{n-1} \cdot X_n) X_n^{\perp}.$$

For (b), note from the previous formulation that $\|\xi_n\|^2 = (V_{n-1} \cdot X_n)^2 \|X_n^{\perp}\|^2 \le \|V_{n-1}\|^2 \|X_n\|^4 / 4$.

Part (c) follows directly from $\mathbb{E}[X_n X_n^T | \mathcal{F}_{n-1}] = A$.

For (d), we use $\|V_n\|^2 = \|V_{n-1} + \gamma_n \xi_n\|^2 = \|V_{n-1}\|^2 + \gamma_n^2 \|\xi_n\|^2 \ge \|V_{n-1}\|^2$. $\qquad\square$

We now check that $(V_n \cdot v^*)^2$ grows in expectation with each iteration.

**Lemma A.2.** *For any $n > n_o$, we have*

    *(a) $(V_n \cdot v^*)^2 \ge (V_{n-1} \cdot v^*)^2 + 2\gamma_n (V_{n-1} \cdot v^*)(\xi_n \cdot v^*)$.*

    *(b) $\mathbb{E}[\xi_n \cdot v^* | \mathcal{F}_{n-1}] = (V_{n-1} \cdot v^*)(\lambda_1 - G(V_{n-1}))$.*

*Proof.* Part (a) follows directly from the update rule:

$$(V_n \cdot v^*)^2 = ((V_{n-1} \cdot v^*) + \gamma_n (\xi_n \cdot v^*))^2 \ge (V_{n-1} \cdot v^*)^2 + 2\gamma_n (V_{n-1} \cdot v^*)(\xi_n \cdot v^*).$$

Part (b) follows by substituting the expression for $\mathbb{E}[\xi_n | \mathcal{F}_{n-1}]$ from Lemma **??**(c):

$$\mathbb{E}[\xi_n \cdot v^* | \mathcal{F}_{n-1}] = (V_{n-1}^T A v^*) - G(V_{n-1})(V_{n-1} \cdot v^*) = \lambda_1 (V_{n-1} \cdot v^*) - G(V_{n-1})(V_{n-1} \cdot v^*).$$

$\qquad\square$

In order to use Lemma **??** to bound the change in potential $\Psi_n$, we need to relate $\Psi_n$ to the quantity $\lambda_1 - G(V_n)$.

**Lemma A.3.** *For any $n \ge n_o$, we have $\lambda_1 - G(V_n) \ge (\lambda_1 - \lambda_2)\Psi_n$.*

*Proof.* It is easiest to think of $V_n$ in the eigenbasis of $A$: the component of $V_n$ in direction $v^*$ is $V_n \cdot v^*$, and the orthogonal component is $V_n^{\perp} = V_n - (V_n \cdot v^*)v^*$. Then

$$G(V_n) = \frac{V_n^T A V_n}{\|V_n\|^2} = \frac{(V_n \cdot v^*)^2}{\|V_n\|^2}\lambda_1 + \frac{(V_n^{\perp})^T A V_n^{\perp}}{\|V_n\|^2} \le \frac{\lambda_1 (V_n \cdot v^*)^2 + \lambda_2 \|V_n^{\perp}\|^2}{\|V_n\|^2}.$$

Therefore,

$$\lambda_1 - G(V_n) \ge \lambda_1 - \frac{\lambda_1 (V_n \cdot v^*)^2 + \lambda_2 (\|V_n\|^2 - (V_n \cdot v^*)^2)}{\|V_n\|^2} = (\lambda_1 - \lambda_2)\left(1 - \frac{(V_n \cdot v^*)^2}{\|V_n\|^2}\right) = (\lambda_1 - \lambda_2)\Psi_n.$$

$\qquad\square$

We can now explicitly bound the expected change in $\Psi_n$ in each iteration.

**Lemma A.4.** *For any $n > n_o$, we can write $\Psi_n \leq \Psi_{n-1} + \beta_n - Z_n$, where $\beta_n = \gamma_n^2 B^2/4$ and where*

$$Z_n = 2\gamma_n(V_{n-1} \cdot v^*)(\xi_n \cdot v^*)/\|V_{n-1}\|^2$$

*is a $\mathcal{F}_n$-measurable random variable with the following properties:*

- $\mathbb{E}[Z_n|\mathcal{F}_{n-1}] = 2\gamma_n(\widehat{V}_{n-1} \cdot v^*)^2(\lambda_1 - G(V_{n-1})) \geq 2\gamma_n(\lambda_1 - \lambda_2)\Psi_{n-1}(1 - \Psi_{n-1}) \geq 0.$

- $|Z_n| \leq 4\gamma_n B.$

*Proof.* Using Lemmas **??** and **??**(a),

$$\Psi_n = \frac{\|V_n\|^2 - (V_n \cdot v^*)^2}{\|V_n\|^2} \leq \frac{\|V_{n-1}\|^2 + \gamma_n^2\|\xi_n\|^2 - (V_n \cdot v^*)^2}{\|V_{n-1}\|^2}$$

$$\leq 1 + \frac{1}{4}\gamma_n^2 B^2 - \frac{(V_n \cdot v^*)^2}{\|V_{n-1}\|^2}$$

$$\leq 1 + \frac{1}{4}\gamma_n^2 B^2 - \frac{(V_{n-1} \cdot v^*)^2 + 2\gamma_n(V_{n-1} \cdot v^*)(\xi_n \cdot v^*)}{\|V_{n-1}\|^2}$$

$$= \Psi_{n-1} + \frac{1}{4}\gamma_n^2 B^2 - 2\gamma_n\frac{(V_{n-1} \cdot v^*)(\xi_n \cdot v^*)}{\|V_{n-1}\|^2},$$

which is $\Psi_{n-1} + \beta_n - Z_n$. The conditional expectation of $Z_n$ can be determined from Lemma **??**(b):

$$\mathbb{E}[Z_n|\mathcal{F}_{n-1}] = \frac{2\gamma_n(V_{n-1} \cdot v^*)}{\|V_{n-1}\|^2}\mathbb{E}[\xi_n \cdot v^*|\mathcal{F}_{n-1}] = 2\gamma_n(\widehat{V}_{n-1} \cdot v^*)^2(\lambda_1 - G(V_{n-1}))$$

and this can be lower-bounded using Lemma **??**.

Finally, we need to determine the range of possible values of $Z_n$. By expanding $\xi_n$, we get

$$Z_n = 2\gamma_n(\widehat{V}_{n-1} \cdot v^*)\left((X_n \cdot v^*)(X_n \cdot \widehat{V}_{n-1}) - (\widehat{V}_{n-1} \cdot v^*)(X_n \cdot \widehat{V}_{n-1})^2\right).$$

Since $\|X_n\|^2 \leq B$, we see that $Z_n$ must lie in the range $\pm 4\gamma_n B$. □

## A.2 The change in potential of the Oja update

Recall the Oja update:

$$V_n = \frac{V_{n-1} + \gamma_n X_n X_n^T V_{n-1}}{\|V_{n-1} + \gamma_n X_n X_n^T V_{n-1}\|}.$$

Since our bounds are on the potential function $\Psi_n$, which is insensitive to the length of $V_n$, we can skip the normalization, and instead just consider the update rule

$$V_n = V_{n-1} + \gamma_n X_n X_n^T V_{n-1}.$$

The final bounds, as well as many of the intermediate results, are almost exactly the same as for Krasulina's estimator. Here is the analogue of Lemma **??**.

**Lemma A.5.** *For any $n > n_o$, we can write $\Psi_n \leq \Psi_{n-1} - Z_n + \beta_n$, where $Z_n$ is the same as in Lemma **??** and $\beta_n = 5\gamma_n^2 B^2 + 2\gamma_n^3 B^3$.*

*Proof.* This is a series of calculations. First,

$$(V_n \cdot v^*)^2 = ((V_{n-1} \cdot v^*) + \gamma_n(V_{n-1}^T X_n X_n^T v^*))^2$$

$$\geq (V_{n-1} \cdot v^*)^2 + 2\gamma_n(V_{n-1} \cdot v^*)(V_{n-1}^T X_n X_n^T v^*).$$

Similarly,

$$\|V_n\|^2 = \|V_{n-1} + \gamma_n X_n X_n^T V_{n-1}\|^2$$

$$= \|V_{n-1}\|^2 + \gamma_n^2\|X_n X_n^T V_{n-1}\|^2 + 2\gamma_n(V_{n-1} \cdot X_n)^2$$

$$\leq \|V_{n-1}\|^2(1 + \gamma_n^2 B^2 + 2\gamma_n(\widehat{V}_{n-1} \cdot X_n)^2)$$

where we have used $\|X_n\|^2 \leq B$. Combining these,

$$
\begin{aligned}
\frac{(V_n \cdot v^*)^2}{\|V_n\|^2} &\geq \frac{(V_{n-1} \cdot v^*)^2 + 2\gamma_n(V_{n-1} \cdot v^*)(V_{n-1}^T X_n X_n^T v^*)}{\|V_{n-1}\|^2(1 + \gamma_n^2 B^2 + 2\gamma_n(\widehat{V}_{n-1} \cdot X_n)^2)} \\
&= \frac{(\widehat{V}_{n-1} \cdot v^*)^2 + 2\gamma_n(\widehat{V}_{n-1} \cdot v^*)(\widehat{V}_{n-1}^T X_n X_n^T v^*)}{1 + \gamma_n^2 B^2 + 2\gamma_n(\widehat{V}_{n-1} \cdot X_n)^2} \\
&\geq \left( (\widehat{V}_{n-1} \cdot v^*)^2 + 2\gamma_n(\widehat{V}_{n-1} \cdot v^*)(\widehat{V}_{n-1}^T X_n X_n^T v^*) \right) \left( 1 - \gamma_n^2 B^2 - 2\gamma_n(\widehat{V}_{n-1} \cdot X_n)^2 \right) \\
&\geq (\widehat{V}_{n-1} \cdot v^*)^2 + 2\gamma_n(\widehat{V}_{n-1} \cdot v^*) \left( \widehat{V}_{n-1}^T X_n X_n^T v^* - (\widehat{V}_{n-1} \cdot v^*)(\widehat{V}_{n-1} \cdot X_n)^2 \right) - 5\gamma_n^2 B^2 - 2\gamma_n^3 B^3
\end{aligned}
$$

where the final step involves some extra algebra that we have omitted. The lemma now follows by invoking $\Psi_n = 1 - (\widehat{V}_n \cdot v^*)^2$. $\qquad\square$

## B  A large deviation bound for $\Psi_n$

### B.1  Proof of Lemma 2.3

For any $t > 0$,

$$
\begin{aligned}
\mathbb{E}\left[ e^{tY_n} | \mathcal{F}_{n-1} \right] &\leq \mathbb{E}\left[ e^{t(Y_{n-1} + \beta_n - Z_n)} | \mathcal{F}_{n-1} \right] \\
&= e^{t(Y_{n-1} + \beta_n)} \mathbb{E}\left[ e^{-tZ_n} | \mathcal{F}_{n-1} \right] \\
&= e^{t(Y_{n-1} + \beta_n)} \mathbb{E}\left[ e^{-t\mathbb{E}[Z_n | \mathcal{F}_{n-1}]} e^{-t(Z_n - \mathbb{E}[Z_n | \mathcal{F}_{n-1}])} | \mathcal{F}_{n-1} \right] \\
&\leq e^{t(Y_{n-1} + \beta_n - \mathbb{E}[Z_n | \mathcal{F}_{n-1}])} \mathbb{E}\left[ e^{-t(Z_n - \mathbb{E}[Z_n | \mathcal{F}_{n-1}])} | \mathcal{F}_{n-1} \right].
\end{aligned}
$$

We bound the last expected value using Hoeffding's lemma: $\mathbb{E}[e^{tW}] \leq e^{t^2(b-a)^2/8}$ for any random variable $W$ of mean zero and range $[a, b]$.

### B.2  Proof of Lemma 2.4

By Lemma 2.3,

$$
\mathbb{E}\left[ e^{tY_n} | \mathcal{F}_{n-1} \right] \leq \exp\left( tY_{n-1} + t\beta_n + \frac{t^2 \zeta_n^2}{8} \right).
$$

Now let's define an appropriate martingale. Let $\tau_n = \sum_{\ell > n}(\beta_\ell + t\zeta_\ell^2/8)$, and let $M_n = \exp(t(Y_n + \tau_n))$. Thus $M_n \in \mathcal{F}_n$, and

$$
\mathbb{E}[M_n | \mathcal{F}_{n-1}] = \mathbb{E}[e^{tY_n} | \mathcal{F}_{n-1}] \exp(t\tau_n) \leq \exp\left( tY_{n-1} + t\beta_n + \frac{t^2 \zeta_n^2}{8} + t\tau_n \right) = M_{n-1}.
$$

Thus $(M_n)$ is a positive-valued supermartingale adapted to $(\mathcal{F}_n)$. A version of Doob's martingale inequality—see, for instance, page 274 of [6]—then says that for any $m$, we have $\Pr(\sup_{n \geq m} M_n \geq \delta) \leq (\mathbb{E}M_m)/\delta$. Using this, we see that for any $\Delta > 0$,

$$
\begin{aligned}
\Pr\left( \sup_{n \geq m} Y_n \geq \Delta \right) &\leq \Pr\left( \sup_{n \geq m} Y_n + \tau_n \geq \Delta \right) = \Pr\left( \sup_{n \geq m} M_n \geq e^{t\Delta} \right) \\
&\leq \frac{\mathbb{E}M_m}{e^{t\Delta}} = \exp(-t(\Delta - \tau_m))\mathbb{E}e^{tY_m}
\end{aligned}
$$

### B.3  Proof of Lemma 2.5

It is well known that $V$ can be chosen by picking $d$ values $Z = (Z_1, \ldots, Z_d)$ independently from the standard normal distribution and then setting $V = Z/\|Z\|$. Therefore,

$$
Y = \frac{Z_2^2 + \cdots + Z_d^2}{Z_1^2 + (Z_2^2 + \cdots + Z_d^2)} = \frac{W_1}{W_1 + W_2},
$$

where $W_1$ is drawn from a chi-squared distribution with $d-1$ degrees of freedom and $W_2$ is drawn independently from a chi-squared distribution with one degree of freedom. This characterization implies that $Y$ follows the Beta$((d-1)/2, 1/2)$ distribution: specifically, for any $0 < y < 1$,

$$\Pr(Y = y) = \frac{\Gamma(\frac{d}{2})}{\Gamma(\frac{d-1}{2})\Gamma(\frac{1}{2})} \, y^{(d-3)/2}(1-y)^{-1/2}.$$

The moment-generating function of this distribution is

$$\mathbb{E}e^{tY} = \frac{\Gamma(\frac{d}{2})}{\Gamma(\frac{d-1}{2})\Gamma(\frac{1}{2})} \int_0^1 e^{ty} y^{(d-3)/2}(1-y)^{-1/2} dy.$$

There isn't a closed form for this, but an upper bound on the integral can be obtained. Assuming $d \geq 3$,

$$\int_0^1 e^{ty} y^{(d-3)/2}(1-y)^{-1/2} dy \leq \int_0^1 e^{ty}(1-y)^{-1/2} dy$$

$$= \frac{e^t}{\sqrt{t}} \int_0^t e^{-z} z^{-1/2} dz$$

$$\leq \frac{e^t}{\sqrt{t}} \int_0^\infty e^{-z} z^{-1/2} dz = \frac{e^t}{\sqrt{t}}\Gamma(1/2),$$

where the second step uses a change of variable $z = t(1-y)$, and the fourth uses the definition of the gamma function. To finish up, we use the inequality $\Gamma(z+1/2) \leq \sqrt{z}\,\Gamma(z)$ (Lemma **??**) to get

$$\mathbb{E}e^{tY} \leq \frac{\Gamma(\frac{d}{2})}{\Gamma(\frac{d-1}{2})} \frac{e^t}{\sqrt{t}} \leq e^t \sqrt{\frac{d-1}{2t}}.$$

The following inequality is doubtless standard; we give a short proof here because we are unable to find a reference.

**Lemma B.1.** *For any $z > 0$,*

$$\Gamma\left(z + \frac{1}{2}\right) \leq \sqrt{z}\,\Gamma(z).$$

*Proof.* Suppose a random variable $T > 0$ is drawn according to the density $\Pr(T = t) \propto t^{z-1}e^{-t}$. Let's compute $\mathbb{E}T$ and $\mathbb{E}\sqrt{T}$:

$$\mathbb{E}T = \frac{\int_0^\infty t^z e^{-t} dt}{\int_0^\infty t^{z-1} e^{-t} dt} = \frac{\Gamma(z+1)}{\Gamma(z)} = z$$

$$\mathbb{E}\sqrt{T} = \frac{\int_0^\infty t^{z-1/2} e^{-t} dt}{\int_0^\infty t^{z-1} e^{-t} dt} = \frac{\Gamma(z+1/2)}{\Gamma(z)},$$

where we have used the standard fact $\Gamma(z+1) = z\Gamma(z)$. By concavity of the square root function, we know that $\mathbb{E}\sqrt{T} \leq \sqrt{\mathbb{E}T}$. This yields the lemma. $\qquad\square$

## B.4 Proof of Theorem 2.2

From Lemma **??**(a), we have $\Psi_n \leq \Psi_{n-1} + \beta_n - Z_n$, where $\beta_n = \gamma_n^2 B^2/4$, and $\mathbb{E}[Z_n|\mathcal{F}_{n-1}] \geq 0$, and $Z_n$ lies in an interval of length $\zeta_n = 8\gamma_n B$. We can thus directly apply the first deviation bound of Lemma 2.4.

Since

$$\sum_{\ell > n} \gamma_n^2 = c^2 \sum_{\ell > n} \frac{1}{\ell^2} \leq c^2 \int_n^\infty \frac{dx}{x^2} = \frac{c^2}{n},$$

we see that for any $t > 0$,

$$\sum_{\ell > n_o} \left( \beta_\ell + \frac{t\zeta_\ell^2}{8} \right) = \sum_{\ell > n_o} \left( \frac{B^2}{4}\gamma_\ell^2 + 8B^2 t\gamma_\ell^2 \right) \leq \frac{B^2 c^2}{4n_o}(1 + 32t).$$

To make this $\leq \epsilon/d$, it suffices to take $n_o \geq B^2 c^2 d(1 + 32t)/(4\epsilon)$, whereupon Lemma 2.4 yields

$$\Pr\left( \sup_{n \geq n_o} \Psi_n \geq 1 - \frac{\epsilon}{d} \right) \leq \mathbb{E}[\exp(t\Psi_{n_o})]e^{-t(1-(\epsilon/d)-(\epsilon/d))}$$

$$\leq e^t \sqrt{\frac{d}{2t}} e^{-t(1-(2\epsilon/d))} = e^{2\epsilon t/d}\sqrt{\frac{d}{2t}}.$$

where the last step uses Lemma 2.5. The result follows by taking $t = d/(4\epsilon)$.

## C  Intermediate epochs of improvement

### C.1  Proof of Lemma 2.7

Lemma **??** establishes an inequality $\Psi_n \leq \Psi_{n-1} - Z_n + \beta_n$ as well as a lower bound on $\mathbb{E}[Z_n|\mathcal{F}_{n-1}]$, where $Z_n$ is a random variable that lies in an interval of length $\zeta_n = 8\gamma_n B$. From Lemma 2.3, we then have

$$\mathbb{E}[e^{t\Psi_n}|\mathcal{F}_{n-1}] \leq \exp\left( t(\Psi_{n-1} - \mathbb{E}[Z_n|\mathcal{F}_{n-1}] + \beta_n + t\zeta_n^2/8) \right)$$
$$\leq \exp\left( t(\Psi_{n-1} - 2\gamma_n(\lambda_1 - \lambda_2)\Psi_{n-1}(1 - \Psi_{n-1}) + \gamma_n^2 B^2(1 + 32t)/4) \right)$$
$$= \exp\left( t(\Psi_{n-1} - c_o\Psi_{n-1}(1 - \Psi_{n-1})/n + c^2 B^2(1 + 32t)/4n^2) \right)$$

For any $\omega \in \Omega_n'$, we have $\Psi_{n-1}(\omega) \leq 1 - \epsilon_j$. Taking expectations over $\Omega_n'$, we get the lemma.

### C.2  Proof of Lemma 2.8

Let $j$ be the largest index such that $n_j < n$. Then

$$\Psi_{n-1}(\omega) \text{ has value } \begin{cases} \leq 1 - \epsilon_j & \text{for } \omega \in \Omega_n' \\ > 1 - \epsilon_j & \text{for } \omega \in \Omega_{n-1}' \setminus \Omega_n' \end{cases}$$

Thus the expected value of $g(\Psi_{n-1})$ over $\Omega_n'$ is at most the expected value over $\Omega_{n-1}'$.

### C.3  Proof of Lemma 2.9

We begin with the following Lemma.

**Lemma C.1.** *For any $n > n_j$ and any $t > 0$,*

$$\mathbb{E}_n[e^{t\Psi_n}] \leq \exp\left( t(1 - \epsilon_j)\left(\frac{n_j + 1}{n + 1}\right)^{c_o\epsilon_j} + \frac{tc^2 B^2(1 + 32t)}{4}\left(\frac{1}{n_j} - \frac{1}{n}\right) \right).$$

*Proof.* Define $\alpha_n = 1 - (c_o\epsilon_j/n)$ and $\xi_n(t) = c^2 B^2 t(1 + 32t)/4n^2$. By Lemmas 2.7 and 2.8, for $n > n_j$,

$$\mathbb{E}_n[e^{t\Psi_n}] \leq \mathbb{E}_n[e^{t\alpha_n\Psi_{n-1}}]\exp(\xi_n(t)) \leq \mathbb{E}_{n-1}[e^{(t\alpha_n)\Psi_{n-1}}]\exp(\xi_n(t)).$$

By applying these inequalities repeatedly, for $n$ shrinking to $n_j + 1$ (and $t$ shrinking as well), we get

$$
\begin{aligned}
\mathbb{E}_n[e^{t\Psi_n}] &\leq \mathbb{E}_{n_j+1}\left[\exp\left(t\Psi_{n_j}\alpha_n\alpha_{n-1}\cdots\alpha_{n_j+1}\right)\right]\exp(\xi_n(t))\exp(\xi_{n-1}(t\alpha_n))\cdots\exp(\xi_{n_j+1}(t\alpha_n\cdots\alpha_{n_j+2})) \\
&\leq \mathbb{E}_{n_j+1}\left[\exp\left(t\Psi_{n_j}\alpha_n\alpha_{n-1}\cdots\alpha_{n_j+1}\right)\right]\exp(\xi_n(t))\exp(\xi_{n-1}(t))\cdots\exp(\xi_{n_j+1}(t)) \\
&= \mathbb{E}_{n_j+1}\left[\exp\left(t\Psi_{n_j}\left(1-\frac{c_o\epsilon_j}{n}\right)\left(1-\frac{c_o\epsilon_j}{n-1}\right)\cdots\left(1-\frac{c_o\epsilon_j}{n_j+1}\right)\right)\right] \times \\
&\quad \exp\left(\frac{c^2B^2t(1+32t)}{4}\left(\frac{1}{n^2}+\frac{1}{(n-1)^2}+\cdots+\frac{1}{(n_j+1)^2}\right)\right) \\
&\leq \exp\left(t(1-\epsilon_j)\exp\left(-c_o\epsilon_j\left(\frac{1}{n_j+1}+\cdots+\frac{1}{n}\right)\right)\right) \times \\
&\quad \exp\left(\frac{c^2B^2t(1+32t)}{4}\left(\frac{1}{n^2}+\frac{1}{(n-1)^2}+\cdots+\frac{1}{(n_j+1)^2}\right)\right)
\end{aligned}
$$

since $\Psi_{n_j}(\omega) \leq 1 - \epsilon_j$ for all $\omega \in \Omega'_{n_j+1}$. We then use the summations

$$
\frac{1}{n_j+1}+\cdots+\frac{1}{n} \geq \int_{n_j+1}^{n+1}\frac{dx}{x} = \ln\frac{n+1}{n_j+1}
$$

$$
\frac{1}{(n_j+1)^2}+\cdots+\frac{1}{n^2} \leq \int_{n_j}^{n}\frac{dx}{x^2} = \frac{1}{n_j}-\frac{1}{n}
$$

to get the lemma. $\qquad\square$

To prove Lemma 2.9, we note that under conditions (3),

$$
(1-\epsilon_j)\left(\frac{n_j+1}{n_{j+1}+1}\right)^{c_o\epsilon_j} \leq e^{-\epsilon_j}(e^{-5/c_o})^{c_o\epsilon_j} = e^{-6\epsilon_j} \leq 1-3\epsilon_j \leq 1-\epsilon_{j+1}-\epsilon_j.
$$

We have used the fact that $e^{-2x} \leq 1-x$ for $0 \leq x \leq 3/4$. The rest follows by applying Lemma **??** with $n = n_{j+1}$.

### C.4 Proof of Lemma 2.10

Pick any $0 < j \leq J$. We will mimic the reasoning of Theorem 2.2, being careful to define martingales only on the restricted space $\Omega'_{n_j}$ and with starting time $n_j$. Then

$$
P_{n_j}\left(\sup_{n\geq n_j}\Psi_n > 1-\epsilon_j\right) \leq \mathbb{E}_{n_j}[e^{t\Psi_{n_j}}]\exp\left(-t(1-\epsilon_j)+\frac{tc^2B^2(1+32t)}{4n_j}\right)
$$

$$
\leq \exp\left(-t\epsilon_{j-1}+\frac{tc^2B^2(1+32t)}{4n_{j-1}}\right),
$$

where the second step invokes Lemma 2.9.

To finish, we pick $t = (2/\epsilon_o)\ln(4/\delta)$. The lower bound on $n_o$ is also a lower bound on $n_{j-1}$, and implies that $tc^2B^2(1+32t)/4n_{j-1} \leq t\epsilon_o/2$, whereupon

$$
P_{n_j}\left(\sup_{n\geq n_j}\Psi_n > 1-\epsilon_j\right) \leq \exp\left(-\frac{t\epsilon_{j-1}}{2}\right) = \left(\frac{\delta}{4}\right)^{\epsilon_{j-1}/\epsilon_o} \leq \frac{\delta}{2^{j+1}}.
$$

Summing over $j$ then yields the lemma.

## D   The final epoch

### D.1   Proof of Lemma 2.11

By Lemma **??**,

$$
\mathbb{E}[\Psi_n|\mathcal{F}_{n-1}] \leq \Psi_{n-1}(1-2\gamma_n(1-\Psi_{n-1})(\lambda_1-\lambda_2))+\beta_n.
$$

For realizations $\omega \in \Omega'_n$, we have $\Psi_{n-1}(\omega) \leq 1/2$ and thus the right-hand side of the above expression is at most $(1 - \alpha_n)\Psi_{n-1} + \beta_n$. Using the fact that $\Omega'_n$ is $\mathcal{F}_{n-1}$-measurable, and taking expectations over $\Omega'_n$,

$$\mathbb{E}_n[\Psi_n] \leq (1 - \alpha_n)\mathbb{E}_n[\Psi_{n-1}] + \beta_n$$
$$\leq (1 - \alpha_n)\mathbb{E}_{n-1}[\Psi_{n-1}] + \beta_n,$$

as claimed. The last step uses Lemma 2.8.

### D.2 Proof of Theorem 1.1

Define epochs $(n_j, \epsilon_j)$ that satisfy the conditions of Theorem 2.6, with $\epsilon_J = 1/2$, and with $\epsilon_{j+1} = 2\epsilon_j$ whenever possible. Then $J = \log_2 1/(2\epsilon_o)$ and

$$n_J + 1 = (n_o + 1)\exp\left(\frac{5J}{c_o}\right) = (n_o + 1)\left(\frac{1}{2\epsilon_o}\right)^{5/(c_o \ln 2)} = (n_o + 1)\left(\frac{4ed}{\delta^2}\right)^{5/(c_o \ln 2)}.$$

By Theorem 2.6, with probability $> 1 - \delta$, we have $\Psi_n \leq 1/2$ for all $n \geq n_J$. More precisely, $P(\Omega'_n) \geq 1 - \delta$ for all $n > n_o$.

By Lemma 2.11, for $n > n_J$,

$$\mathbb{E}_n[\Psi_n] \leq \left(1 - \frac{a}{n}\right)\mathbb{E}_{n-1}[\Psi_{n-1}] + \frac{b}{n^2},$$

for $a = c_o/2$ and $b = c^2 B^2/4$. By the $a > 1$ case of Lemma ??,

$$\mathbb{E}_n[\Psi_n] \leq \left(\frac{n_J + 1}{n + 1}\right)^a \mathbb{E}_{n_J}[\Psi_{n_J}] + \frac{b}{a - 1}\left(1 + \frac{1}{n_J + 1}\right)^{a+1}\frac{1}{n + 1}$$
$$\leq \frac{1}{2}\left(\frac{n_o + 1}{n + 1}\right)^a\left(\frac{4ed}{\delta^2}\right)^{5/(2\ln 2)} + \frac{b}{a - 1}\exp\left(\frac{a + 1}{n_J + 1}\right)\frac{1}{n + 1}.$$

which upon further simplification yields the bound of Theorem 1.1 for $a > 1$.

(Note that the $a < 1$ case of Lemma ?? yields a rate of $\mathbb{E}_n[\Psi_n] = O(n^{-a})$.)

**Lemma D.1.** *Consider a nonnegative sequence $(u_t : t \geq t_o)$, such that for some constants $a, b > 0$ and for all $t > t_o \geq 0$,*

$$u_t \leq \left(1 - \frac{a}{t}\right)u_{t-1} + \frac{b}{t^2}.$$

*Then, writing the zeta function $\zeta(s) = \sum_{i=1}^{\infty} i^{-s}$,*

$$u_t \leq \begin{cases} \left(\frac{t_o+1}{t+1}\right)^a u_{t_o} + \frac{b}{a-1}\left(1 + \frac{1}{t_o+1}\right)^{a+1}\frac{1}{t+1} & , \quad a > 1 \\ \\ \left(\frac{t_o+1}{t+1}\right)^a u_{t_o} + 4b\zeta(a-2)\frac{1}{(t+1)^a} & , \quad a < 1 \end{cases}$$

*Proof.* Recursively applying the given recurrence for $u_t$ yields

$$u_t \leq \left(\prod_{i=t_o+1}^{t}\left(1 - \frac{a}{i}\right)\right)u_{t_o} + \sum_{i=t_o+1}^{t}\frac{b}{i^2}\left(\prod_{j=i+1}^{t}\left(1 - \frac{a}{j}\right)\right).$$

To bound the product term, we use

$$\prod_{i=t_o+1}^{t}\left(1 - \frac{a}{i}\right) \leq \exp\left(-a\sum_{i=t_o}\frac{1}{i}\right) \leq \exp\left(-a\int_{t_o+1}^{t+1}\frac{dx}{x}\right) = \left(\frac{t_o + 1}{t + 1}\right)^a.$$

Therefore,

$$u_t \leq \left(\frac{t_o+1}{t+1}\right)^a u_{t_o} + \sum_{i=t_o+1}^{t} \frac{b}{i^2}\left(\frac{i+1}{t+1}\right)^a$$

$$\leq \left(\frac{t_o+1}{t+1}\right)^a u_{t_o} + \frac{b}{(t+1)^a}\left(\frac{t_o+2}{t_o+1}\right)^2 \sum_{i=t_o+1}^{t}(i+1)^{a-2}.$$

We finish by bounding the summation of $(i+1)^{a-2}$ by a definite integral, to get:

$$\sum_{i=t_o+1}^{t}(i+1)^{a-2} \leq \begin{cases} \frac{1}{a-1}(t+2)^{a-1} & , \ a > 1 \\ \zeta(a-2) & , \ a < 1 \end{cases}.$$

$\square$