[Reviews · NeurIPS 2013]

Submitted by Assigned_Reviewer_7

This paper proves fast convergence rates for Oja's well-known incremental algorithm for PCA. The proof uses a novel technique to describe the progress of the algorithm, by breaking it into several "epochs"; this is necessary because the PCA problem is not convex, and has saddle points. The proof also uses some ideas from the study of stochastic gradient descent algorithms for strongly convex functions. The theoretical bounds give some insight into the practical performance of Oja's algorithm, and its sensitivity to different parameter settings.

Quality: The results of this paper are nice, and the proof is technically quite impressive.

Some minor comments: The paper claims a convergence rate of O(1/n), but to avoid confusion, one should note that this is in terms of a potential function \Psi_n that is roughly the *square* of the L2 distance between the algorithm's output and the true solution.

Also, in section 4, the authors assume B*c=1 without loss of generality because the data can be rescaled; but is this really true? Rescaling the data changes B, but it also changes the preferred setting for c, namely c = 1/(2(\lambda_1-\lambda_2)), as described on line 177. It seems like B*c = constant, but not necessarily 1.

Clarity: The paper is clearly written. The proof overview in section 5.1 is helpful.

Minor comments: on line 368, "Fig. 3" should be "Fig. 4"; in Fig. 5, the graph should be labeled "Dependence on h," not "Dependence on c."

Originality: The ideas used in this paper seem quite original; it is always a bit surprising when a simple algorithm provably solves a non-convex problem.

Significance: The results are great from a theoretical standpoint, and have some practical significance as well. (Oja's algorithm may not perform as well as more sophisticated methods, but it is simple and easy to use, so having a better understanding of its performance is useful.)
Summary: A strong paper, with interesting theoretical analysis of a practically-relevant algorithm.

Submitted by Assigned_Reviewer_8

This paper theoretically analyse the Oja’s algorithm for estimating the first principal component. Convergence properties and relation to stochastic gradient descent have been discussed. Overall, this paper is not well written and should not be accepted in the current form.

Detail comments are:

1. All theoretical analyses are based on the potential functions defined in line 109-111. There is no citation for this potential function and it is not explained why this function is defined in this form. Using the norm of the difference between v_n and the optimal first principal component is a more standard way to analyse the convergence rate. The author should explain why it is not adopted in this paper.

2. The assumption of the example in section 3 is too strong. It is inappropriate to assume all data points are drawn from eigenvectors of the covariance matrix. It is only a special case which cannot be used as a fact to analyse the initialization situation and the further theoretical proofs in Section 4 and Section 5.

3. In line 153-155, “if v_n is orthogonal to some e_i, it will remain so forever” is not true, because x_n^Tv_{n-1} is not always 0 and v_n is not always equals to v_{n-1}. x_n^Tv_{n-1} = 0 only when x_n is e_i. However, x_n could be e_j, where j\neq i, according to the construction of the data points.

4. The submitted version is not compiled properly which has many question marks when citing Appendix. The supplementary file has the right citation.
However, the correctness of proofs is still hard to be verified because of the extremely poor organization of the structure. For example:
The proof of Theorem 4.1 ->A1->Theorem 5.2->E.1->C.1->F.1-F.3,
where “->” means “refers to”. It is impossible to follow unless using a clear structure.

5. This paper cites too many arxiv papers. Most of these papers have been published on peer reviewed journals or conferences. Please cite the peer reviewed version.

6. There are many grammar mistakes which make the paper difficult to read.
For example, the section paragraph in Introduction is not readable.
Summary: This paper is not well prepared. It may have some interesting observations. However, they are not clearly shown in this paper.

Submitted by Assigned_Reviewer_9

This paper provides a theoretical justification for the statistical performance of incremental PCA. They prove the \tilde{O}(1/n) finite sample rate of convergence for estimating the leading eigenvector of the covaraince matrix. Their results suggest the best learning rate for incremental PCA. Also, their analysis provide insights for relationship with SGD on strongly convex functions. I find the result of this paper very solid. but there are some issues to be addressed.

1. I think the author should clarify the gap between their result and Kuzmin's result. Though the author claims "though such guarantees are more general than ours", it is better to clearly state the detailed difference since the result of the result of this paper may be not novel given Kuzmin's result.

2. It seems strange to me that the high probability result in Theorem 4.2 doesn't depends on the dimension d. Why the upper bound of \Psi_n doesn't depend on d? The dependency on d is only in the lower bound of n. Also, the author needs to explain the implication of the P(.) defined on the nested sample space. Is this result weaker than the common high probability results? In sum, this result might be correct, but it should be presented in a more illustrative way.

3. I find the relation to strongly convex stochastic gradient descent very insightful.

4. I think the experiment is not presented in an illustrative way. Comparison between the two initialization ways are helpful.
Summary: This paper provides a theoretical justification for the statistical performance of incremental PCA. The results are very interesting but have to be presented in better ways. Also the experiments need to be improved.
Author Feedback

Author rebuttal: We thank all the reviewers for their insightful comments!


Assigned_Reviewer_7:

- The squared L2 norm potential function is standard in the gradient descent/first-order optimization literature, where O(1/n) is claimed in such situations; we will make a note of this and fix the other typos you noted.
- We agree with your comment on the value of Bc. As you noted, the value of c needed to ensure fast O(1/n) convergence would change with rescaled data, and our analysis is readily redone to depend on Bc. The algorithm would not change, though, and the solution it would find is invariant to global rescaling; so if c is not high enough for fast convergence, B could in principle be reduced and the algorithm run again. We therefore regarded the B-dependence as less fundamental and omitted it for clarity.



Assigned_Reviewer_8:

Addressing your points:
1) Our potential function reduces exactly to the standard potential function ||v_n - v^*||^2 when h=1, since v_n and v^* are unit length (squared L2 norm is standard in analyses of gradient descent and other first-order optimization methods). We therefore strictly generalize the standard one to look at convergence to higher principal subspaces. We will explicitly note this in the paper.

2) We seek a universal initialization method that would work for any distribution in order to make the problem concrete, since neither the algorithm nor our guarantees assume a specific data-generating distribution. The example given in Section 3 is a lower bound showing that the intuitively appealing method of initializing v_0 to be the first example will not work for some distributions. This is our justification for using the uniform distribution over the sphere as the initialization for v_0. We believe there could be better initialization methods that immediately take advantage of low intrinsic dimension (e.g. a randomly perturbed data point). Overall, our main result concerns the rate of convergence and not the initialization method.

3) The statement is correct as stated, but the way it is written might be confusing. Here is a hopefully less confusing explanation. The distribution is concentrated on the unit vectors +/- e_1, +/- e_2,.... Therefore the initial vector v_0 is one of these unit vectors; call it e_i. For any future examples x_n=+/- e_j there are two possibilities:
* i \neq j: in which case v_n \cdot x_n = 0 and therefor v_{n+1}=v_n
* i = j: in which case v_{n+1} = c e_i for some scalar c
In either case, v_n is equal to c e_i forever.

4) We agree that the organization of the proofs can be improved and will improve it in the revised paper, possibly by combining sections.
5) We agree with these comments. We will replace the Arxiv references with the appropriate conference/journal references.
6) We will do our best to improve the grammar. Specific pointers to the locations of the mistakes would be helpful.



Assigned_Reviewer_9:

Addressing your points:
1) Kuzmin et al. analyze a worst-case (adversarial) model, whereas we consider stochastic optimization in which data points are from an a-priori-fixed distribution, which is essentially strictly “easier” in terms of guarantees (see Cesa-Bianchi et al.'s 2004 work relating the two settings). Their algorithm has basically the same time complexity every iteration as PCA on the whole dataset, though, so it makes no sense for our setting. Our guarantees are for a simple linear-time algorithm every iteration, which is the novelty here.

2) We agree that the results can be presented more illustratively, and will clear up the statements of the main results. There are two specific questions you asked here:
- The upper bound of \Psi_n does indeed depend on d through the "constants" \alpha_i in Thm. 4.2; those are only constants with respect to n, since we wanted to highlight the n-dependence of the guarantees (we believe these are surprising for such a non-convex problem). We will similarly present the expected value results to reflect this. The dependence on d could doubtless be improved, but is not our focus (though Section 3 provides a useful target dependence).
- The high-probability result basically states as long as n is high enough, there is a high-probability event over the initialization (\Omega_n) within which the usual (strongly convex SGD) high-probability result holds. As such, it is strictly slightly weaker than strongly convex SGD results. This appears to be the price of nonconvexity (all the saddle points) of the objective function. Nonconvex optimization methods often rely on random restart schemes to escape local minima; our result shows how a simple algorithm without restarts can provably converge fast with high probability over a single initialization.

4) We agree that the experiments could be more extensive. Our focus was to simply illustrate the effect on convergence of the c-value and of the eigenvalue spectrum; both effects match our theory's predictions in detail, as the experiments indicate. We did not compare the initialization methods for space reasons, because in practice they perform very similarly. The algorithm depends heavily on the eigenspectrum in converging to higher and higher principal eigenspaces, and this seems to quickly dominate initialization effects. Also, the focus of our work is this simple algorithm’s fast convergence (with n) to principal eigenspaces and the implications on parameter setting of c, not the initialization method. We will explicitly convey this in the paper.